# Circulating miRNA-451a and miRNA-328-3p as Potential Markers of Coronary Artery Aneurysmal Disease

**DOI:** 10.3390/ijms24065817

**Published:** 2023-03-18

**Authors:** Sylwia Iwańczyk, Tomasz Lehmann, Artur Cieślewicz, Katarzyna Malesza, Patrycja Woźniak, Agnieszka Hertel, Grzegorz Krupka, Paweł P. Jagodziński, Marek Grygier, Maciej Lesiak, Aleksander Araszkiewicz

**Affiliations:** 11st Department of Cardiology, Poznan University of Medical Sciences, 61-848 Poznań, Poland; 2Department of Biochemistry and Molecular Biology, Poznan University of Medical Sciences, 60-781 Poznań, Poland; 3Clinical Pharmacology, Poznan University of Medical Sciences, 61-848 Poznań, Poland

**Keywords:** microRNAs, coronary aneurysm, coronary artery disease, atherosclerosis

## Abstract

MicroRNAs (miRNAs) are currently investigated as crucial regulatory factors which may serve as a potential therapeutic target. Reports on the role of miRNA in patients with coronary artery aneurysmal disease (CAAD) are limited. The present analysis aims to confirm the differences in the expression of previously preselected miRNAs in larger study groups and evaluate their usefulness as potential markers of CAAD. The study cohort included 35 consecutive patients with CAAD (Group 1), and two groups of 35 patients matched Group 1 regarding sex and age from the overall cohort of 250 patients (Group 2 and Group 3). Group 2 included patients with angiographically documented coronary artery disease (CAD), while Group 3 enrolled patients with normal coronary arteries (NCA) assessed during coronary angiography. We applied the RT-qPCR method using the custom plates for the RT-qPCR array. We confirmed that the level of five preselected circulating miRNAs was different in patients with CAAD compared to Group 2 and Group 3. We found that miR-451a and miR-328 significantly improved the CAAD prediction. In conclusion, miR-451a is a significant marker of CAAD compared to patients with CAD. In turn, miR-328-3p is a significant marker of CAAD compared to patients with NCA.

## 1. Introduction

Circulating non-coding RNAs have been proposed as a new class of biomarkers and are investigated for their potential in many areas of cardiology, including coronary artery disease (CAD) [1]. As critical regulatory factors, microRNAs (miRNAs) may serve as a potential therapeutic target [2,3]. Reports on the role of miRNA in patients with coronary artery aneurysmal disease (CAAD) are limited and relate mainly to patients with Kawasaki disease.

CAAD is a rare condition affecting epicardial arteries. It occurs in 1.2% to 4.9% of patients undergoing coronary angiography [4]. CAAD includes two phenotypes: coronary artery aneurysm (CAA) and coronary artery ectasia (CAE) [5]. CAA is defined as a focal dilatation of the artery with a diameter of 1.5 times the adjacent normal coronary artery, whereas CAE describes more diffuse dilatation, often involving the predominant part of the artery and more than one vessel [6]. Moreover, CAA is classified into fusiform type if the longitudinal dimension is greater than the transverse or saccular type when the transverse dimension exceeds the longitudinal [7].

In symptomatic patients, the most common clinical presentation of CAAD is stable angina pectoris. It is significantly less frequent that myocardial infarction (MI) and aneurysm complications occur, i.e., thrombus formation following distal embolization, development of a fistula, and aneurysm rupture [6,7,8]. Moreover, CAAD is associated with poor long-term outcomes, regardless of concomitant atherosclerotic coronary disease [9,10].

The most common cause of abnormal dilatation of the coronary arteries is the atherosclerotic process followed by positive remodeling of the vessel. Approximately 20 to 30% of CAADs are considered inflammatory or congenital [11,12]. However, pathogenesis is multifactorial and can be influenced by the disturbance of physiological processes at the molecular level. Understanding these pathomechanisms is crucial for diagnosing and preventing CAAD complications. According to a growing body of literature, miRNAs significantly regulate human physiological processes, including gene expression in the cardiovascular system [13].

Based on our previous study [14], we selected the potential miRNA markers to validate them in a larger independent patient cohort. The present analysis aims to confirm the differences in the expression of five preselected miRNA markers (miR-125b-5p, miR-210-3p, miR-328-3p, miR-425-3p, and miR-483-5p), and three reference miRNAs (miR-16-5p, miR-30d-5p, miR-320d) in larger study groups and evaluate their usefulness as potential markers of CAAD occurrence.

## 2. Results

### 2.1. Clinical Characteristics of the Study Population

We presented baseline clinical characteristics in Table 1. Patients were predominantly male at the mean age of 65.9 ± 9.4, 67.1 ± 8.3, and 64.6 ± 12.3 in Group 1, Group 2, and Group 3, respectively. The prevalence of major cardiovascular risk factors did not differ significantly between the study groups. Despite the comparable incidence of renal failure, serum creatinine levels were significantly higher in Groups 1 and 2 compared to Group 3. In addition, Group 3 patients had no history of myocardial infarction (MI) or percutaneous coronary intervention (PCI). Therefore, a significantly lower group percentage was treated with aspirin and P2Y12 inhibitor. Moreover, beta-blocker and statin use were significantly less frequent in Group 3 than in Group 2. There were no differences in pharmacotherapy between Groups 1 and 2.

### 2.2. Expression Profile of miRNAs in the Plasma of Group 1 Versus Group 2 and Group 3

We confirmed that the concentration of five preselected plasma miRNAs was different in patients with CAAD (Group 1) compared to patients with CAD (Group 2) and patients without significant coronary artery disease (Group 3) (Table 2). We identified a specific signature of plasma miRNAs up- or down-regulated in individual groups.

### 2.3. Comparison of Group 1 with Group 2

The univariate logistic regression analysis revealed three independent variables potentially increasing the risk of CAAD (Table 3).

We included all of them in the multivariate logistic regression analysis. Model 1 included two independent clinical variables, previous percutaneous coronary intervention (PCI) and dyslipidemia. Despite the low model fit (R2Pseudo = 0.11), it was statistically significant (*p* = 0.004). Both previous PCI and dyslipidemia significantly reduced the risk of CAAD. In Model 2, miR-451a improved the model fit (R2Pseudo = 0.16) and was associated with a significant reduction of the risk of CAAD (Table 3).

Comparative analysis of Model 1 and Model 2 based on R2Pseudo and the Likelihood-Ratio Test (LRT) (*p* = 0.038) revealed a significantly better fit of Model 2, including miR-451a, than of Model 1 (AIC = 83.6, AIC = 85.9, for Model 2 and Model 1, respectively) (Figure 1).

### 2.4. Comparison between Group 1 and Group 3

The univariate logistic regression analysis revealed six independent clinical variables and five miRNAs potentially increasing the risk of CAAD (Table 4). We included all of them in the multivariate logistic regression analysis.

Model 1 included all independent clinical variables, such as previous myocardial infarction (MI), P2Y12 inhibitor and aspirin taking, white blood cell (WBC) count, red blood cell (RBC) count, and serum creatinine level. It was significantly associated with the risk of CAAD compared to patients without significant coronary artery lesions (*p* < 0.01). In addition, the goodness-of-fit was quite high (R2Pseudo = 0.46). Among the independent variables included in the model, only taking the P2Y12 inhibitor significantly increased the risk of CAAD (OR [95% CI] 80.3 [5.8; 1108.4, *p* < 0.01). In Model 2, miRNAs were added separately to Model 1. Only miR-328-3p improved the model fit (R2Pseudo = 0.62) and was associated with a significant reduction in the risk of CAAD (OR [95% CI] 0.21 [0.07; 0.67], *p* = 0.01) (Table 5).

Comparative analysis of Model 1 and Model 2 based on R2Pseudo and LRT (*p* < 0.01) revealed a significantly better fit of Model 2, including miR-328-3p, than of Model 1 (AIC = 49.0, AIC = 61.4, for Model 2 and Model 1 respectively) (Figure 2).

### 2.5. Functional Enrichment Analysis

Recent research reports that exogenous, circulating miRNAs can enter recipient cells and play a regulatory role in cells or organisms [15]. Assuming that miRNA is significantly deregulated in CAAD and CAD samples, and may be transported into epithelial, smooth muscle, or myocyte cells, we undertook in silico functional enrichment analysis. To identify biochemical signaling pathways potentially affected by miRNA modulated in CAAD/CAD patients, we analyzed the predicted target genes of the five significantly affected miRNAs (hsa-miR-451a, hsa-miR-23a-3p, hsa-miR-210-3p, hsa-miR-328-3p, hsa-miR-425-3p, hsa-miR-483-5p) (Table 6). The target genes of the five identified miRNAs were extracted from the miRWalk database (Appendix A), and the intersection of identified target genes from GSEA tool programs was chosen [15,16]. We correlated predicted target genes with the KEGG (Kyoto Encyclopedia of Genes and Genomes) to identify potential biochemical pathways as the functional output of five deregulated miRNAs. The enrichment analysis represents pathways significantly enriched with target genes for miRNAs in CAAD and CAD patients in relation to the control group. Significantly (FDR < 0.05) enriched KEGG pathways are presented in Table 6. Annotated notes about predicted target genes for the five dysregulated miRNAs in CAA and CAD groups are presented in Appendix A.

The KEGG enrichment analysis indicated that the five miRNAs targeted genes are involved in ten pathways. The AMPK signaling pathway is involved in vast processes in the heart [17]: e.g., modulation of vascular inflammation and vascular smooth muscle cells phenotype [18]. Several miR-23-3p potential targets, in the AMPK pathway, were selected: TSC1, PIK3R1, PPARGC1A, FOXO3, and one miR-328-3p target SCD.

Endothelial cells, vascular smooth muscle cells, and macrophages play important roles in the development of CAD. In the etiology of atherosclerosis, these cells undergo two types of programmed cell death apoptosis and autophagy [19,20]. Numerous studies have revealed recently that autophagy-related genes (ATGs) are involved in endothelial and smooth-muscle inappropriate autophagy in the processes of atherogenesis [21]. In our in silico analysis, several genes were selected as potential targets for miRNAs. The following miRNAs were selected as regulators of autophagy-related genes: miR-328-3p HMGB1, miR-23-3p TSC1, PIK3R1, miR-210-3p ATG7, and VMP1. An example is HMGB1, the intranuclear non-histone DNA binding protein, which contributes to the stabilization of nucleosomes and the development of atherosclerosis [22].

## 3. Discussion

Coronary artery aneurysmal disease (CAAD) is a progressive, gradual coronary artery abnormality that mainly results in recurrent angina or, less commonly, severe complications such as distal embolization and myocardial infarction. The critical factor that initiates and deteriorates the course of CAAD is atherosclerosis. Asymptomatic course and progression make an early diagnosis of CAAD difficult. miRNAs play a role in the mechanism of atherosclerosis, and atherosclerotic coronary artery disease is characterized by specific signatures of miRNA in tissue and plasma assays [23,24,25,26]. However, scarce data concerning miRNAs involved in CAAD pathology prompted us to study five circulating miRNAs: MiR-125b-5p, miR-210-3p, miR-328-3p, miR-425-3p, and miR-483-5p, which we preselected and described in our former study [14] as potentially associated with CAAD.

This study evaluated the predictive role of circulating miRNAs for CAAD. Through a sampling process, we found miR-451a and miR-328 to be the most effective for CAAD prediction in a general Greater Poland population. Adding miR-451a and miR-328-3p to traditional risk factors significantly improved the detection of CAAD. We found that a higher circulating level of miR-328-3p was significantly associated with a higher risk of CAAD than the control group. On the other hand, the higher plasma level of miR-451a was predictive of the higher risk of CAAD compared to CAD.

As described in the methodology, miRNAs appeared as stable molecules and were recognized as an advantage of good potential markers for cardiovascular issues [27]. Selecting miRNA markers from dozens of circulating miRNA requires a multistep procedure. We overcame this difficulty and applied RT-qPCR to select miRNA markers for CAAD.

Since the presence of miRNAs in plasma was first mentioned in the literature, circulating miRNAs have been reported as a potential biomarker of heart failure and CAD [23,28]. In contrast to the numerous studies in CAD cohorts, the predictive value of miRNAs in CAAD is yet to be discovered. CAAD is usually detected during coronary angiography. Although data suggest a few CAD biomarkers, the need for reliable CAAD biomarkers, discriminating it from CAD, remains a limitation in the clinical practice. Our study suggested miRNA as a minimally invasive biomarker to improve pre-procedure diagnosis of CAAD. We applied the RT-qPCR method using a PCR array system. The custom plates for the RT-qPCR array were designed as in our former study [14].

We confirmed that the concentration of five preselected plasma miRNAs was different in patients with CAAD (Group 1) compared to patients with ischemic heart disease (Group 2) and patients without significant coronary artery disease (Group 3).

By applying logistic regression, we compared two models without and with miRNA. The model without miRNA to compare CAAD (Group 1) and CAD (Group 2) included two independent variables: previous percutaneous coronary intervention (PCI) and dyslipidemia. By adding miR-451a to the model, the prediction increased. miR-451a appeared to be a significant marker of CAAD compared to the group of patients with CAD.

The model without miRNA to compare CAAD (Group 1) and the control (Group 3) included six independent variables, previous MI, P2Y12 inhibitor or aspirin taking, WBC count, RBC count, and serum creatinine level. By adding miR-328-3p to the model, the prediction increased. miR-328-3p appeared to be a significant marker of CAAD compared to patients without significant CAD.

The available papers concerning miRNA as a marker of CAAD are scarce. RT-qPCR evaluated let-7i-3p as a biomarker to identify CAAD patients [29]. Serum exosomal let-7i-3p was proposed as a diagnostic marker for CAAD in patients with Kawasaki disease [29]. miRNAs miR-451a and miR-328, which we elaborated on in the present study, have already been proposed as potential markers of heart failure (HF) and CAD.

miR-451a expression is elevated in the plasma of patients with acute myocardial infarction (AMI) compared with unstable angina pectoris and healthy control groups [30]. Data suggest that miR-451a regulates ischemic heart injury [31]. Overexpression of miR-451a in mice significantly attenuated Ang II-induced cardiac fibrosis and inflammation. miR-451a directly targeted transcription factor TBX1 expression, thus decreasing TBX1-dependent TGFB1 gene expression. In our study, we observed a higher plasma level of miR-451a in CAAD patients compared to CAD patients. Therefore, we conclude that miR-451a could be involved in CAAD pathomechanism since CAAD is a severe consequence of coronary vessel inflammation and atherosclerosis. Recently, miR-451a has been also found to inhibit the expression of both MMP-2 and MMP-9 in cardiomyocytes by macrophage migration inhibitory factor (MIF) [32].

In our study, miR-328 was increased in the CAAD group compared to the control group. An increased miR-328 plasma level was reported in patients with AMI, strongly associated with increased risk of mortality or heart failure [33]. Elevated expression of miR-328 is also a marker of cardiac fibrosis during MI. This observation has recently been demonstrated in vitro and explained by the paracrine mechanism. miR-328 is secreted by cardiomyocytes promoting fibrosis by regulating adjacent fibroblasts [34]. Transgenic mice with miR-328 overexpression in the heart induced cardiac hypertrophy [35]. miR-328 reduces the transcript of sarco/endoplasmic reticulum Ca^2+^-ATPase (SERCA2a, *ATP2A2*), a transporter of Ca^2+^ from the cytosol into the sarcoplasmic reticulum [35,36,37].

miR-23a, miR-210, and miR-425 were significantly regulated in the CAAD group versus the control group, but did not significantly influence a logistic regression model improving prediction. Our study demonstrated decreased miR-23a in the CAAD group versus the control. Previous studies suggested increased plasma miR-23a in CAD patients [38,39]. It has been suggested to be used to predict the presence and severity of coronary lesions in patients with CAD. In our study, miR-23a has not been significantly modulated in the CAD group.

So far, several studies evaluating the prognostic value of single circulating miRNAs in cardiovascular disease have been reported [40]. miR-210 derived from peripheral blood predict mortality and represent a valuable biomarker for risk estimation in CAD [41,42]. Circulating miR-210 improved the prediction model, including AUC (area under the receiver-operating characteristic curve). In our study, miR-210 was increased in CAAD vs. control but did not improve the model, including clinical parameters.

miRbase miR-425-3p is a minor transcript in mammalian cells [43]. Several studies demonstrated that predominant miR-425-p functions as a negative regulator of cardiac fibrosis by suppressing *TGFB1* (encoding TGFβ1) expression [44].

Our finding shows a new look at the miRNA signatures for plasma samples from CAAD patients. It provides a composed clinical and molecular predictive model, benefiting the diagnosis of CAAD patients. Developing a method using miRNAs as CAAD markers could improve the early diagnosis of CAAD and identify patients at increased risk of progression and complications.

We proposed a new model of CAAD prediction using miR-451a and miR-328 in addition to independent clinical variables. Our approach with logistic regression models revealed a significant increase in the predictability of discrimination between the CAAD group and the CAD group, as well as the CAAD group against the control group, by adding the status of miR-451a and miR-328, respectively.. This preclinical assessment of potential miRNA markers could be developed in the future in reliable tests for early diagnosis of CAAD.

Our achievement has limits in the number of patients compared to the study. According to several studies, the condition, time of the day, and several other parameters could influence miRNA levels and should be controlled and standardized in future studies.

In future endeavours is the question of how miR-451a and miR-328 contribute to CAAD development. The important question also concerns target cells for miRNA and the source of abnormal levels of selected markers in plasma.

## 4. Materials and Methods

### 4.1. Study Design and Patient Selection

The overall cohort enrolled two hundred fifty patients undergoing coronary angiography for angina symptoms between April 2020 and December 2021. The study cohort included 35 consecutive patients with CAAD (Group 1), and two groups of 35 patients matched Group 1 in terms of sex and age from the overall cohort (Group 2 and Group 3) (Figure 3). Group 2 included patients with angiographically documented CAD, while Group 3 enrolled patients with normal coronary arteries (NCA) assessed during coronary angiography. Due to hemolysis in the two CAAD patient samples, we finally analyzed 33 patients.

All patients were qualified for coronary angiography, according to the European Society of Cardiology (ESC) guidelines [45]. According to the available data, CAAD is a diffuse or focal dilatation of the coronary artery with a diameter of 1.5 times the adjacent normal segment. We also enrolled patients with concomitant coronary artery stenosis in Group 1. The angiographic criteria for CAD included: coronary artery stenosis > 90% or intermediate stenosis (50–90%) with documented ischemia or hemodynamically significant, defined as either fractional flow reserve (FFR) ≤ 0.80 or an instantaneous wave-free ratio (iFR) ≤ 0.89. Patients included in Group 3 had normal ECG and echocardiography apart from no significant lesions in the coronary arteries. In addition, they showed no evidence of ischemia during non-invasive exercise testing.

Patients meeting the following criteria were excluded from the study: (i) acute coronary syndrome; (ii) elevated levels of myocardial injury markers (troponin I or creatine kinase); (iii) severe hepatic and renal dysfunction present or history; (iv) hematological disease; (v) severe inflammatory and malignant diseases; (vi) systemic connective tissue disease; (vii) biological treatment; and (viii) no informed consent.

### 4.2. Blood Samples Collection and Evaluation of miRNA Profiles

Our previous study precisely described obtaining and preparing blood samples [14]. We treated the new collection of blood samples according to the same procedure described here. We collected blood samples (10 mL) from patients on the first day after the coronary angiography and processed them within 30 min. Samples were centrifuged at 1300× *g* for 15 min at room temperature. The supernatant was transferred to RNase-free tubes and then stored at −80 °C.

The miRNAs were isolated from individual 200 μL frozen plasma using the miRNeasy Serum/Plasma Advanced Kit (Cat. No 217204, Qiagen, Dusseldorf, Germany) according to the manufacturer’s instructions. To correct sample-to-sample variation, a synthetic set of spike-ins, UniSp2, UniSp4, and UniSp5 was applied (RNA Spike-In Kit for reverse transcription (RT) Cat. No 339390, Qiagen, Dusseldorf, Germany). Three of the RNA spike-in templates (UniSp2, UniSp4, and UniSp5) were provided premixed in one vial, each at a different concentration in 100-fold increments (UniSp2 (2 fmol/μL), UniSp4 (0.02 fmol/μL), and UniSp5 (0.0002 fmol/μL)). Before starting the RNA isolation procedure, one microliter of this RNA spike-in mix per RNA prepared with 60 µL Lysis Buffer RPL was aliquoted. Approximate RNA quantity and quality were estimated using a NanoDrop 2000 spectrophotometer (Thermo Scientific, Waltham, MA, USA). Four microliters of each total RNA were reversely transcribed with the miRCURY LNA RT Kit (Cat. No 339340, Qiagen, Dusseldorf, Germany). The UniSp6 RNA spike-in from the miRCURY LNA RT Kit and cel-miR-39-3p RNA from miRCURY LNA RT Kit were applied as controls.

Twenty microliters of cDNA product of the miRCURY LNA RT Kit were applied for each quantitative PCR (qPCR). qPCR was performed using miRCURY LNA SYBR Green PCR Kit (Cat. No 339345, 339346, 339347, Qiagen, Dusseldorf, Germany) with miRCURY LNA Custom PCR Panels (Qiagen) (Table 7).

Table 8 summarizes the genes analyzed in the custom plate, including preselected miRNA markers and spike-in controls applied in the analysis: the control, hemolysis markers, and reference markers. We provided a detailed description of the spike-in controls in the appendix (Appendix B).

### 4.3. Statistical Analysis

We presented all continuous variables as means with standard deviation for normal distribution or medians (upper and lower quartile) for non-normal distribution. We tested the normality of the distribution of variables using the Kolmogorov–Smirnov test and presented categorical variables as counts and percentages or frequencies. We assessed the significance of differences between the mean values of the continuous data consistent with the normal distribution using one-way ANOVA and Tukey’s Test. To compare the continuous data inconsistent with the normal distribution, we used Mann–Whitney and Kruskal–Wallis tests with Benjamini–Hochberg post hoc analysis and compared categorical variables using the χ^2^ test. We performed an analysis of array data with GENEGLOBE online software (https://geneglobe.qiagen.com/pl/analyze, accessed on 12 January 2023). We used the global mean as a reference for the PCR-array analysis and calculated the *p* values based on a Student’s *t*-test of replicate 2^(−ΔCT)^ values for each miRNA in the control and treatment groups. Subsequently, we based the *p*-value calculation on a parametric, two-sample equal variance, unpaired, two-tailed analysis. In the current analysis, the selection criteria to elaborate potential miRNA markers were the fold-regulation ≥ 2 and *p* ≤ 0.05.

We conducted the univariate logistic regression analysis to evaluate confounding factors for each CAAD comparison (Group 1 vs. Group 2 and Group 1 vs. Group 3). The confounding factor was selected based on a *p* value < 0.1. The multivariate logistic regression analysis was conducted for each comparison with confounding factors that met the criteria in univariate analysis (Model 1). The following step was to create the Model 2 by adding the analyzed miRNA to Model 1, which had an expression that was significantly different between the compared groups (Table 2). We assessed the model fit for Model 1 and Model 2 using the Akaike information criterion (AIC) and pseudo-R-squared values. The model with the lowest AIC value was considered the best. The higher pseudo-R-squared value indicated a better model fit. All statistical analyses were two-sided and were conducted with PQStat Software (PQStat v.1.8.0.476, Poznań, Poland). A *p* value < 0.05 was considered a statistical significance.

## 5. Conclusions

The clinical use of the studied miRNAs as markers requires validation on a bigger cohort or advanced sequencing techniques. However, preliminary analysis indicates that miR-451a is a significant marker of CAAD compared to patients without concomitant aneurysms. In turn, miR-328-3p is a significant marker of CAAD compared to patients without significant changes in the coronary arteries.

## Figures and Tables

**Figure 1 ijms-24-05817-f001:**
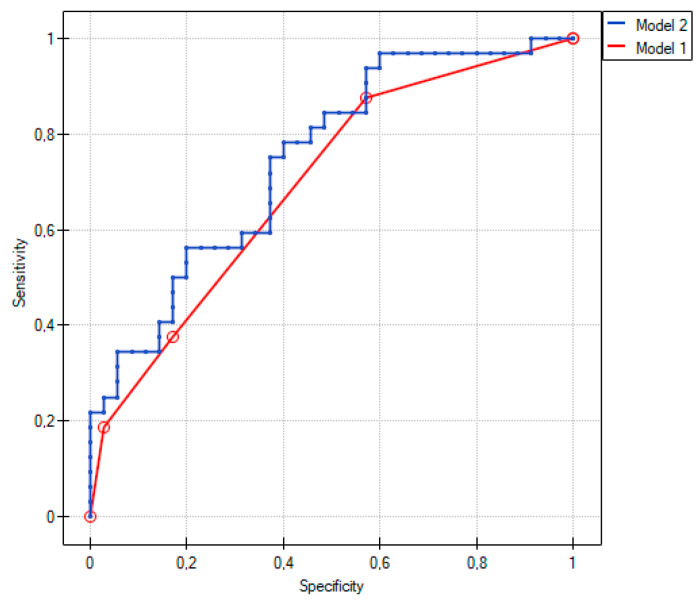
Comparison of logistic regression models between Group 1 and Group 2.

**Figure 2 ijms-24-05817-f002:**
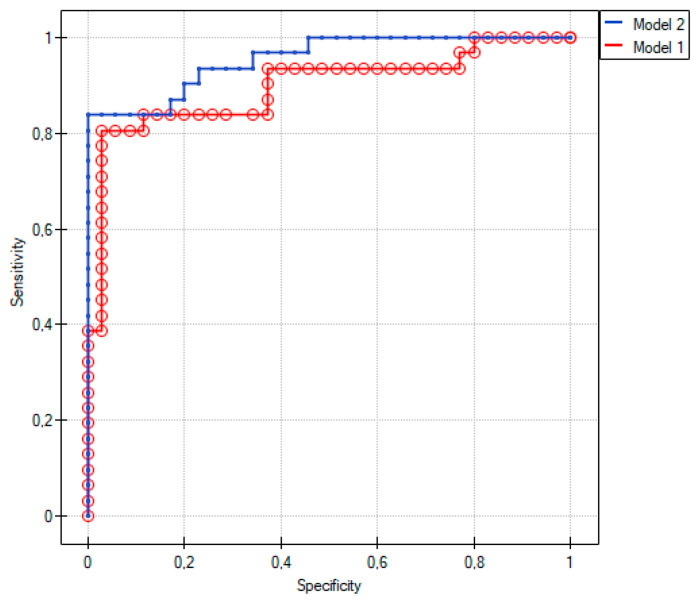
Evaluation of logistic regression models to compare Group 1 and Group 3.

**Figure 3 ijms-24-05817-f003:**
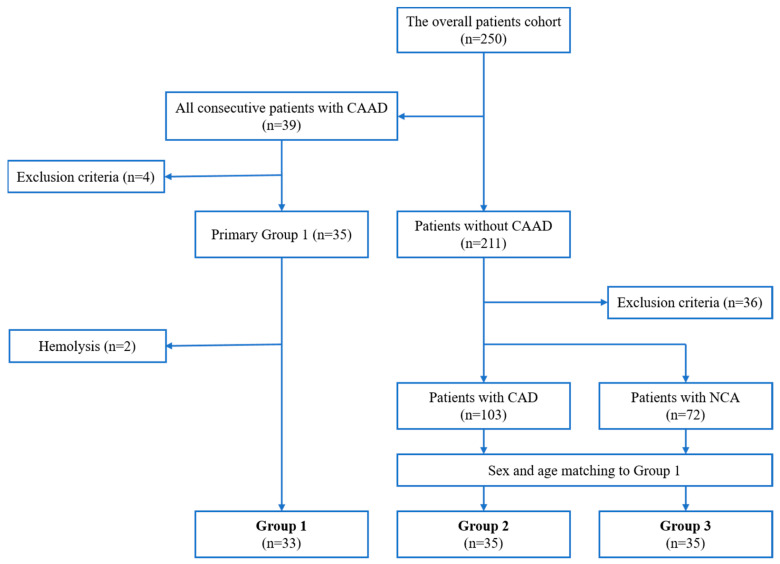
Flowchart of the study population. CAAD, coronary artery aneurysmal disease; CAD, coronary artery disease; NCA, normal coronary arteries.

**Table 1 ijms-24-05817-t001:** Baseline clinical characteristics.

Variable	Group 1 (n = 33)	Group 2 (n = 35)	Group 3 (n = 35)	*p*-Value		
				Group 1 vs. Group 2	Group 1 vs. Group 3	Group 2 vs. Group 3
Sex, male	28 (84.8)	28 (80.0)	27 (77.1)	0.41	0.69	0.18
Age, years	65.9 ± 9.4	67.1 ± 8.3	64.6 ± 12.3	0.88	0.85	0.55
BMI, kg/m^2^	29.7 (26.5–32.9)	28.9 (25.3–31.9)	30.1 (27.2–33.7)	0.72	0.72	0.63
Hypertension	29 (87.9)	30 (85.7)	31 (88.6)	0.81	0.90	0.72
Hyperlipidemia	20 (60.6)	29 (82.9)	27 (77.1)	0.09	0.26	0.55
Diabetes mellitus	10 (30.3)	10 (28.6)	11 (31.4)	0.97	0.77	0.79
HF, NYHA ≥ 2	6 (18.2)	9 (25.7)	4 (11.4)	0.12	0.40	0.12
LVEF, %	57.5 (55–60)	60 (50–60)	60 (55–60)	0.87	0.87	0.87
Stroke	4 (12.1)	3 (8.6)	1 (2.9)	0.90	0.30	0.60
Smoking	11 (33.3)	11 (31.4)	8 (22.8)	0.80	0.30	0.42
CKD, eGFR ≤ 60 mL/min	2 (6.1)	7 (20)	9 (25.7)	0.45	0.88	0.20
Previous MI	10 (33.3)	13 (37.1)	0	0.61	0.01	<0.01
Previous PCI	10 (33.3)	20 (57.1)	0	0.03	<0.01	<0.01
Previous CABG	1 (3.0)	3 (8.6)	0	0.67	0.46	0.24
Drugs administration						
Aspirin	25 (75.8)	26 (74.3)	22 (62.9)	0.60	0.01	0.05
P2Y12 inhibitor	14 (42.4)	20 (57.1)	1 (2.9)	0.24	<0.01	<0.01
Beta-blocker	23 (69.7)	25 (71.4)	22 (62.9)	0.84	0.15	0.04
Statin	26 (78.8)	28 (80.0)	30 (85.7)	0.95	0.12	0.02
ACEI/ARB	25 (75.8)	26 (74.3)	31 (88.6)	0.78	0.82	1.00
OHA	7 (21.2)	6 (17.1)	12 (34.3)	0.96	0.33	0.29
Insulin	2 (6.1)	1 (2.9)	3 (8.6)	0.90	0.74	1.00
Laboratory test						
WBC, 10^9^/L	7.2 (5.9–9.1)	7.5 (6.1–9.6)	6.5 (5.5–7.7)	0.58	0.15	0.05
HGB, mmol/L	9.2 (8.3–9.9)	9 (8.1–9.5)	9 (8.4–9.8)	0.45	0.85	0.45
PLT, 10^9^/L	223.5 (188.5–257)	204 (180–239)	201 (176.5–249.5)	0.91	0.91	0.91
LDL-C, mmol/L	2.1 (1.6–2.6)	1.9 (1.6–3.2)	2.8 (2.0–3.3)	0.78	0.33	0.44
Creatinine, mmol/L	85.7 (77.6–99.5)	93 (80–99.5)	80.9 (69.4–86.0)	0.45	0.04	<0.01
Glucose, mmol/L	5.6 (5.1–6.3)	5.8 (5.2–6.6)	5.8 (5.3–6.4)	0.86	0.86	0.86

We presented continuous variables as a mean ± s.d. or median (interquartile range), and categorical variables as a number (percentage). Abbreviations: ACEI, angiotensin-converting enzyme inhibitor; ARB, angiotensin II receptor blocker; BMI, body mass index; CABG, coronary artery bypass graft; CCB, calcium channel blocker; CKD, chronic kidney disease; HGB, hemoglobin; HF, heart failure; LDL-C, low-density lipoprotein cholesterol; LVEF, left ventricle ejection fraction; MI, myocardial infarction; NYHA, New York Heart Association; OHA, oral hypoglycemic agents; PCI, percutaneous coronary intervention; PLT, platelet count; s.d., standard deviation; WBC, white blood cells; significant difference = *p* < 0.05.

**Table 2 ijms-24-05817-t002:** Deregulated miRNAs in the circulation of all studied groups.

Symbol	Up-Down Regulation					
	Group 2 Compared to Group 1		Group 3 Compared to Group 1		Group 3 Compared to Group 2	
	Fold Regulation	*p*-Value	Fold Regulation	*p*-Value	Fold Regulation	*p*-Value
hsa-miR-451a	−2.18	0.03	−2.64	0.01	−1.51	0.39
hsa-miR-23a-3p	1.54	0.01	2.11	<0.01	1.10	0.59
hsa-miR-125b-5p	−1.51	0.28	−1.78	0.07	−1.51	0.14
hsa-miR-210-3p	−2.07	0.22	−4.71	<0.01	−1.50	0.03
hsa-miR-328-3p	−1.08	0.53	−2.25	0.03	−1.88	0.03
hsa-miR-425-3p	−1.96	0.17	−3.30	0.01	−1.63	0.14
hsa-miR-483-5p	−1.28	0.45	−1.66	0.30	−1.47	0.02
hsa-miR-16-5p	−1.34	0.07	−1.23	0.15	−1.15	0.49
hsa-miR-30d-5p	−1.18	0.96	−1.49	0.15	−1.59	0.18
hsa-miR-320d	−1.38	0.27	−1.46	0.04	−1.38	0.10

The calculation does not include data for the spike-in controls (UniSP2-6 and cel-miR-39-3p) or any blank spots. MiRNA markers selected for further analysis are marked in red.

**Table 3 ijms-24-05817-t003:** The univariate and multivariate logistic regression analysis of clinical variables and miRNA expression profile in Group 1 compared to Group 2.

Variable	Univariate Analysis		Multivariate Analysis	
	OR [95% CI]	*p*-Value	OR [95% CI]	*p*-Value
Previous PCI	0.3 [0.1; 0.9]	0.03	0.2 [0.1; 0.7]	0.01
Dyslipidemia	0.2 [0.1; 1.1]	0.06	0.2 [0.1; 0.8]	0.02
hsa-miR-451a	0.7 [0.5; 0.9]	0.03	0.7 [0.5; 0.99]	0.049

Abbreviations: OR, odds ratio; PCI, percutaneous coronary intervention.

**Table 4 ijms-24-05817-t004:** The univariate logistic regression analysis of clinical variables and miRNA expression profile in Group 1 compared to Group 3.

Variable	Univariate Analysis	
	OR [95% CI]	*p*-Value
Previous MI	7.5 [1.5; 37.6]	0.01
P2Y12 inhibitor	36.3 [4.4; 229.0]	<0.01
Aspirin	5.5 [1.4; 21.8]	0.01
WBC, 10^9^/L	1.2 [0.9; 1.6]	0.06
RBC, 10^9^/L	3.2 [1.1; 10.0]	0.04
Creatinine, mmol/L	1.02 [1.0; 1.1]	0.03
hsa-miR-328-3p	0.7 [0.5; 0.9]	<0.01
hsa-miR-451a	0.6 [0.4; 0.9]	<0.01
hsa-miR-23a-3p	2.6 [1.5; 4.7]	<0.01
hsa-miR-210-3p	0.7 [0.5; 0.8]	<0.01
hsa-miR-425-3p	0.8 [0.6; 0.9]	0.01

Abbreviations: MI, myocardial infarction; RBC, red blood cells; WBC, white blood cells; OR, odds ratio.

**Table 5 ijms-24-05817-t005:** The multivariate logistic regression analysis of clinical variables and miRNA expression profile in Group 1 compared to Group 3 (Model 2).

Variable	Multivariate Analysis	
	OR [95% CI]	*p*-Value
Previous MI	7.0 [0.9; 51.7]	0.06
P2Y12 inhibitor	80.3 [5.8; 1108.4]	<0.01
Aspirin	3.7 [0.4; 29.8]	0.22
WBC, 10^9^/L	1.3 [0.9; 1.9]	0.08
RBC, 10^9^/L	4.5 [0.8; 24.6]	0.11
Creatinine, mmol/L	1.0 [0.97; 1.1]	0.54
hsa-miR-328-3p	0.2 [0.1; 0.7]	<0.01

Abbreviations: MI, myocardial infarction; RBC, red blood cells; WBC, white blood cells; OR, odds ratio.

**Table 6 ijms-24-05817-t006:** Pathways regulated by miRNAs hsa-miR-451a, hsa-miR-23a-3p, hsa-miR-210-3p, hsa-miR-328-3p, hsa-miR-425-3p. The target genes of the five identified miRNAs were extracted from the miRWalk database, and the intersection of identified target genes from GSEA tool programs was chosen. We correlated identified target genes with the KEGG (Kyoto Encyclopedia of Genes and Genomes) to point to biochemical pathways as potentially functional outputs of five deregulated miRNAs.

Pathways	Genes	*p*-Value, Adjusted	*p*-Value (BH)
Non-alcoholic fatty liver disease (NAFLD)	IL6R NDUFV3 PIK3R1 SDHD UQCRFS RAC1	<0.01	0.01
AMPK signaling pathway	TSC1, PIK3R1 PPARGC1A FOXO3 SCD	<0.01	0.01
Autophagy	TSC1 ATG7 PIK3R1 HMGB1 VMP1	<0.01	<0.01
Regulation of actin cytoskeleton	VAV3 EZR PIK3R1 MYH10 RAC1	0.02	0.06
Thermogenesis	TSC1 NDUFV3 SDHD PPARGC1A UQCRFS1	0.03	0.06
Huntington disease	NDUFV SDHD PPARGC1A ACTR1A UQCRFS1	0.05	0.08
PI3K-Akt signaling pathway	TSC1 IL6R PIK3R1 FOXO3 RAC1	0.12	0.18
Pathways in cancer	MITF IL6R TPM3 RUNX1T1 PIK3R1 RAC1	0.20	0.25
Herpes simplex virus 1 infection	TSC1 ZNF275 ZNF268 ZNF701 PIK3R1	0.29	0.32
Metabolic pathways	NDUFV3 HMGCS1 GLS SDHD STT3A MGAT5 FUT4 SCD UQCRFS1 STT3B	0.62	0.62

**Table 7 ijms-24-05817-t007:** miRCURY LNA Custom PCR Panels (Qiagen) primers on a 96-well plate. Columns 1 and 2 were repeated six times. Six samples were analyzed on a single 96-well plate.

	1	2
**A**	UniSp2	hsa-miR-125b-5p
**B**	UniSp4	hsa-miR-210-3p
**C**	UniSp5	hsa-miR-328-3p
**D**	UniSp6	hsa-miR-425-3p
**E**	cel-miR-39-3p	hsa-miR-483-5p
**F**	UniSp3	hsa-miR-16-5p
**G**	hsa-miR-451a	hsa-miR-30d-5p
**H**	hsa-miR-23a-3p	hsa-miR-320d

**Table 8 ijms-24-05817-t008:** Genes analyzed on the Custom plate.

Position	Mature ID	Meaning of the Gene
1	UniSp2	Control
2	UniSp4	Control
3	UniSp5	Control
4	UniSp6	Control
5	cel-miR-39-3p	Control
6	UniSp3	Control
7	hsa-miR-451a	Hemolysis marker
8	hsa-miR-23a-3p	Hemolysis marker
9	hsa-miR-125b-5p	Marker
10	hsa-miR-210-3p	Marker
11	hsa-miR-328-3p	Marker
12	hsa-miR-425-3p	Marker
13	hsa-miR-483-5p	Marker
14	hsa-miR-16-5p	Reference marker
15	hsa-miR-30d-5p	Reference marker
16	hsa-miR-320d	Reference marker

## Data Availability

The data presented in this study are available on request from the corresponding author. The data are not publicly available due to confidential genetic data.

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
