# Peer review of "Circulating miRNA-451a and miRNA-328-3p as Potential Markers of Coronary Artery Aneurysmal Disease"

_ijms, 2023, doi:10.3390/ijms24065817_

Round 1

Reviewer 1 Report

1. miR-451a is a significant marker of CAAD compared to patients with CAD without  concomitant aneurysms. In turn, miR-328-3p is a significant marker of CAAD compared  to patients without significant changes in the coronary arteries. This is a good sign. However, as an important marker for determining whether CAAD is based on its detection value, the accuracy rate should be above 80% in different stages and under different medication environments, and CAAD, as an important marker of a disease, had better be specific. Therefore, in order to achieve clinical application, many comprehensive and in-depth research work needs to be carried out.

2. Lines 382-397 appear to be reviewer comments.

Author Response

Dear Reviewer,

We are grateful for peer-reviewing and recommendations concerning our manuscript entitled: “Circulating miRNA-451a and miRNA-328-3p as potential markers of coronary artery aneurysmal disease.” We appreciate the Reviewer’s effort very much and hope that the answers contained below will be satisfactory

  1. miR-451a is a significant marker of CAAD compared to patients with CAD without concomitant aneurysms. In turn, miR-328-3p is a significant marker of CAAD compared to patients without significant changes in the coronary arteries. This is a good sign. However, as an important marker for determining whether CAAD is based on its detection value, the accuracy rate should be above 80% in different stages and under different medication environments, and CAAD, as an important marker of a disease, had better be specific. Therefore, in order to achieve clinical application, many comprehensive and in-depth research work needs to be carried out.

Response: Thank you for your comment. We agree that clinical use requires further in-depth research work, including confirmation on a larger study group. In the next step, we also plan to analyze the genes encoding selected miRNAs. According to your comment, we modified the conclusion. 

  1. Lines 382-397 appear to be reviewer comments.

Response: Thank you and at the same time we apologize for the oversight. This is a fragment of the template that we did not remove when completing the manuscript. We have already removed the unnecessary part.

Reviewer 2 Report

the manuscript is well structured and very interesting. Diagnosing diseases with miRNAs represents the future of medicine. I have only one question. Is there an expression correlation between mRNAs expressed during myocardial infarction and CAAD? Are they the same mrna or different? I recommend citing this manuscript DOI: 10.3390/diagnostics11010032

Author Response

Dear Reviewer,

We are grateful for peer-reviewing and recommendations concerning our manuscript entitled: “Circulating miRNA-451a and miRNA-328-3p as potential markers of coronary artery aneurysmal disease.” We appreciate the Reviewer’s effort very much and hope that the answers contained below will be satisfactory

The manuscript is well structured and very interesting. Diagnosing diseases with miRNAs represents the future of medicine. I have only one question. Is there an expression correlation between mRNAs expressed during myocardial infarction and CAAD? Are they the same mrna or different? I recommend citing this manuscript DOI: 10.3390/diagnostics11010032

Response: Thank you very much for your comment.

Regarding your question, there is no correlation between having a heart attack and having CAAD (HR 0.77, CI [0.28; 2.12]; p=0.61). There was no statistically significant difference in miRNA expression.

Thanks for the suggestion on literature. We've added it to the references.

Reviewer 3 Report

1. p-value follow the same pattern overall either use 0.01 pattern throughout and follow that or 0.001 and strictly follow that use exponential if values are greater.

2. Patients number is too much small as it is qRT-PCR. at least number should be 50 in each group is must 

3. discussion section needed to be improved, too much focus on the target genes as worked on miRNAs not on target genes.

4. conclusion required validation on a bigger cohort, the author needs to present the conclusion that these miRNAs after validation on a bigger cohort or advanced sequencing techniques can be used as markers. 

Author Response

Dear Reviewer,

We are grateful for peer-reviewing and recommendations concerning our manuscript entitled: “Circulating miRNA-451a and miRNA-328-3p as potential markers of coronary artery aneurysmal disease.” We appreciate the Reviewer’s effort very much and hope that the answers contained below will be satisfactory

  1. p-value follow the same pattern overall either use 0.01 pattern throughout and follow that or 0.001 and strictly follow that use exponential if values are greater.

Response: Thank you for your remark. We corrected the p-value using 0.01 pattern.

  1. Patients number is too much small as it is qRT-PCR. at least number should be 50 in each group is must.

Response: We agree that 35 samples are too small for RT-qPCR testing of reliable, beyond-doubt markers that could be diagnostically useful and significant. We plan to confirm this data on a greater number of samples. According to the Materials and methods section, we explained that groups of patients were earnestly selected from more prominent groups to obtain these well-defined narrow groups strictly fitting the CAAD criteria.

  1. Discussion section needed to be improved, too much focus on the target genes as worked on miRNAs not on target genes.

Response: We agreed with the Reviewer, and we modified the Discussion by eliminating redundant information concerning target genes. However, we left a description of several target genes to show potential links between miRNA and CAAD mechanism, to prove that selected miRNA could play a role in the pathomechanism of CAAD. It remains to be further studied to show what is the involvement of miRNA in CAAD development.

  1. conclusion required validation on a bigger cohort, the author needs to present the conclusion that these miRNAs after validation on a bigger cohort or advanced sequencing techniques can be used as markers.

Response: Thank you for your suggestion. We modified the conclusions according to your comment.

“The clinical use of the studied miRNAs as markers requires validation on a bigger cohort or advanced sequencing techniques. However, preliminary analysis indicates that miR-451a is a significant marker of CAAD compared to patients with CAD without concomitant aneurysms. In turn, miR-328-3p is a significant marker of CAAD compared to patients without significant changes in the coronary arteries.”